# Expression Analyses in the Rachis Hint towards Major Cell Wall Modifications in Grape Clusters Showing Berry Shrivel Symptoms

**DOI:** 10.3390/plants11162159

**Published:** 2022-08-19

**Authors:** Stefania Savoi, Suriyan Supapvanich, Heinrich Hildebrand, Nancy Stralis-Pavese, Astrid Forneck, David P. Kreil, Michaela Griesser

**Affiliations:** 1Department of Agricultural, Forest and Food Sciences, University of Turin, 10095 Grugliasco, Italy; 2Department of Agricultural Education, School of Industrial Education and Technology, King Mongkut’s Institute of Technology Ladkrabang, 1 Chalongkrung Road, Ladkrabang, Bangkok 10520, Thailand; 3Institute of Viticulture and Pomology, Department of Crop Sciences, University of Natural Resources and Life Sciences, Vienna, 3430 Tulln an der Donau, Austria; 4Institute of Molecular Biotechnology, Department of Biotechnology, University of Natural Resources and Life Sciences, Vienna, Muthgasse 18, 1190 Vienna, Austria; 5Institute of Computational Biology, Department of Biotechnology, University of Natural Resources and Life Sciences, Vienna, Muthgasse 18, 1190 Vienna, Austria

**Keywords:** *Vitis vinifera*, gene expression, ripening disorder, cell wall

## Abstract

Berry shrivel (BS) is one of the prominent and still unresolved ripening physiological disorders in grapevine. The causes of BS are unclear, and previous studies focused on the berry metabolism or histological studies, including cell viability staining in the rachis and berries of BS clusters. Herein, we studied the transcriptional modulation induced by BS in the rachis of pre-symptomatic and symptomatic clusters with a custom-made microarray qPCR in relation to a previous RNASeq study of BS berries. Gene set analysis of transcript expression in symptomatic rachis tissue determined suppression of cell wall biosynthesis, which could also be confirmed already in pre-symptomatic BS rachis by CESA8 qPCR analyses, while in BS berries, a high number of SWITCH genes were suppressed at veraison. Additionally, genes associated with the cell wall were differently affected by BS in berries. A high percentage of hydrolytic enzymes were induced in BS grapes in rachis and berries, while other groups such as, e.g., xyloglucan endotransglucosylase/hydrolase, were suppressed in BS rachis. In conclusion, we propose that modulated cell wall biosynthesis and cell wall assembly in pre-symptomatic BS rachis have potential consequences for cell wall strength and lead to a forced degradation of cell walls in symptomatic grape clusters. The similarity to sugar starvation transcriptional profiles provides a link to BS berries, which are low in sugar accumulation. However, further studies remain necessary to investigate the temporal and spatial coordination in both tissues.

## 1. Introduction

Grape berry ripening has been studied intensively over the last decades due to the high economic importance of table and wine grapes worldwide and the rising challenges the viticulture community is facing due to climate change [1,2,3,4]. Thereby, the biochemical and molecular processes, resulting in the typical double sigmoid growth curve of grape berries with three distinct phases [5], have been investigated with modern techniques [6,7,8]. These helped to elucidate the primary [9,10,11,12] and the secondary metabolism [13], including different aroma compounds [14], towards the effects of abiotic stress, e.g., drought [15,16] or heat stress [17,18], as well as changed profiles due to pathogen infections [19]. Attention has also been given to the processes of ripening control via phytohormones [20,21] and the succession of steps necessary for the metabolic shift from symplast to apoplast phloem unloading [22], berry softening [23], and the accumulation and metabolism of primary and secondary compounds in different berry tissues [24,25]. The onset of ripening is characterized by an initial fall of berry elasticity and turgor pressure before ABA signals are observed, and berries start accumulating sugars and producing anthocyanins in the skins [26]. This process seems to be coordinated by a hierarchy of transcriptional signals [7]. Berry softening and the phases of berry growth are controlled by the coordinated expression of cell wall modification enzymes, aquaporin channels, and sugar transporters [27,28].

These enzymes and proteins interplay during berry softening with the cell wall, which confers rigidity to the plant cells. However, controlled modifications of a sophisticated network of cellulose, hemicellulose, and pectin offer enough flexibility to ensure cell growth and expansion [29]. Microfibrils of cellulose (long repeated glucose units joined with β,1-4 glycosidic bonds) and hemicellulose (short cross-linking polysaccharide chains of mainly xylan, xyloglucan, arabinoxylan, glucomannan) are embedded in a matrix of polysaccharides, glycoproteins, proteoglycans, low-molecular-mass compounds, and ions. The main type of polysaccharide present in this matrix is pectin, which is constituted by the polymers homogalacturonan (HGA, a linear homopolymer of galacturonic acid whose molecules are joined by an α-1,4 bond, partially methylesterified acetylated or xylosylated), rhamnogalacturonan I (RG-I, with a backbone composed of the disaccharide (1-2)α-L-rhamnose-(1-4)α-D-galacturonic acid and arabinan and galactan as side chains), and rhamnogalacturonan II (RG-II, a homogalacturonan backbone with four highly-conserved branching chains with borate diester cross-links) [29,30]. Through an iterative calculation, the relative molar distribution (mol%) of the different polysaccharides in grape skins was estimated as 57−62 mol% homogalacturonan, 6.0−14 mol% cellulose, 10−11 mol% xyloglucan, 7 mol% arabinan, 4.5−5.0 mol% rhamnogalacturonan I, 3.5−4.0 mol% rhamnogalacturonan II, 3 mol% arabinogalactan, and 0.5−1.0 mol% mannans [31]. Interestingly, no major compositional changes in cell wall polysaccharides have been determined during grape berry ripening, although modifications of specific components were observed, together with large changes in protein composition [32,33].

In contrast to the well-studied berry metabolism, the development and functions of the rachis of grape clusters and berry pedicels are much less understood. Previous studies focused either on the genetic and environmental plasticity of cluster architecture and compactness, a trait important for grape phytopathology [34,35,36] or on the vascular system conductivity of ripening grape berries, finding distinct functions of phloem and xylem [37,38,39], which are specific tissues supporting the transport of nutrients and assimilated toward berries also by the activity of transporters [40,41]. Interestingly, most studies analyzed the rachis either as part of the post-harvest storage of table grapes [42] or in association with grapevine physiological ripening disorders [43,44,45,46].

Sunburn, late-season dehydration (LSD), bunch stem necrosis (BSN), and an early sugar accumulation disorder called berry shrivel (BS) are among the grapevine ripening disorders that substantially impact grape yield and berry quality. The latest, being the focus of the presented study, is characterized by a stop in sugar accumulation short after veraison, enhanced contents of organic acids, low pH values, and in red grape varieties, reduced biosynthesis of anthocyanins in berry skins [47,48,49].

The Austrian red grape variety Blauer Zweigelt could be highly affected by BS, with incidences between 5 and 40% of affected grapes within one vineyard per year [47,50]. The causes of BS induction remain unclear, but recent studies on both tissues, berries as well as rachis, gave new insights into the complex structural and biochemical modifications associated with BS symptom development. Recently, a transcriptomic approach confirmed early changes in gene expression in grape berries at veraison before visible BS symptoms, but no changes were obtained in samples collected before veraison [49]. Among these early changes in BS berries, a group of so-called SWITCH genes indicates that BS could be due to a delay in ripening. A disturbed grape berry ripening or a delayed ripening induction is one of the hypotheses of berry shrivel induction but results also hint toward a disturbed transport of assimilated and nutrients towards berries in association with reduced cell viability in the rachis and berries [43,45,46,51,52]. Cell death in berries and rachis of BS symptomatic grape clusters have been observed [43,53], and recently it was shown that cell death in the berries precedes the ones in the rachis [46]. Collapsed cells as well as a localized thickening of cell walls in the secondary phloem have been observed on the cellular level in the rachis and pedicels of BS grape clusters [45]. These dramatic changes point toward processes involving cell wall modification and degradation in either BS induction, BS symptoms development, or both. In our previous transcriptomics analysis [48,49,54], we focused on the grape berries to possibly decipher the causes leading to the berry shrivel disorder without finding conclusive answers to the issue. Here, we analyze in detail the transcriptional response in the rachis before and after berry shrivel symptoms and correlate the changes in cell wall metabolism to the one observed in the berry on the transcriptional level. We aim to understand the contribution of these cell wall modifications to BS induction and symptom development. Our results could link the altered metabolism in different tissues in BS gapes and may contribute to finding potential causes of BS.

## 2. Results

### 2.1. Gene Expression Modulation in the Rachis of BS-Affected Grape Clusters

The rachis tissue was collected from pre-symptomatic pre-veraison grape clusters (T1) and symptomatic grape clusters during ripening (T2). These time points are of interest for learning about (the still unknown) genes affected in the rachis that can lead to the berry shrivel phenotype or reflect its consequences.

For each of the two sampling dates, we tested genes for strong effects in BS relative to healthy controls (H). In pre-symptomatic and pre-veraison samples (T1), only 39 genes were significantly more than 2-fold differentially expressed, and all of them were enhanced in BS rachis (|log_2_[fold change]| > 1, Appendix A). GeneOntology (GO) functional enrichment analysis revealed that several cell-wall-related categories such as the “xyloglucan metabolic process”, the “cell wall organization or biogenesis”, the “glucan metabolic process”, the “hemicellulose metabolic process”, etc., were significantly over-represented, despite the small numbers of genes tested (*q* < 5% false discovery rate, Benjamini–Hochberg correction; Figure 1).

With the ripening disorder’s progression and visible symptoms of berry shrivel, a higher number of genes were affected in T2; in fact, 427 genes were significantly more than 2-fold differentially expressed (|log_2_[fold change]| > 1), of which 192 were enhanced and 235 repressed (Appendix A). Furthermore, the GO functional analysis of the 427 genes showed an over-representation of cell-wall-related categories, indicating an involvement of cell wall metabolism and of cell wall structure in symptomatic rachis tissues (Figure 1). Moreover, several hydrolytic enzymes were highly induced, such as β,1-3-glucanase and β-galactosidase; in contrast, highly repressed genes included cell wall modification enzymes such as the xyloglucan endotransglycosylases and several cellulose synthases, part of the cellulose biosynthesis group (Appendix A). Genes with strong differential expression are summarized in Table 1, grouped according to the categories “cell wall biosynthesis”, “cell wall degradation”, “cell wall modification”, and “hydrolytic enzymes of polysaccharides”. We observed lower expression in genes involved in cellulose synthase, pectinesterase family, and xyloglucan endotransglucosylase/hydrolases at T2. In contrast, among the highest up-regulated genes observed are BXL1 (*VIT_05s0077g01280*)—a β-xylosidase, XTH32 (*VIT_06s0061g00550*)—a xyloglucan endotransglucosylase/hydrolase, EXPA6 (*VIT_06s0004g04860*)—an expansin, and genes coding for hydrolytic enzymes of polysaccharides: β-galactosidase (*VIT_11s0016g02200*) and β-1,3-glucanase (*VIT_08s0007g06040*, *VIT_06s0061g00120*).

To complement GeneOntology enrichment analyses, we performed a gene set analysis (GSA) for a selection of gene sets of interest. Comparing all berry shrivel samples pairwise to their healthy controls showed lower expression of genes involved in “cell wall biosynthesis cellulose synthesis” (score −0.590, Benjamini–Hochberg-corrected FDR *p*-value 0.015). In parallel, cell wall modifying and hydrolytic genes were enhanced: “cell wall modification expansin” (0.649, *p*-value 0.07), “hydrolytic enzymes endo-1,3-β-glucosidase” (0.322, *p*-value 0.07), “hydrolytic enzyme gluco-, galacto- and mannosidases” (0.232, *p*-value 0.08), “cell wall degradation pectate lyases and polygalacturonases” (0.623, *p*-value 0.08) (Appendix A).

Samples for a higher-resolution time course were collected at six sampling dates in 2013 from EL-32 till EL-36/2, covering the first phase of berry growth (green, hard berries), veraison (EL-35) towards ripening berries (fully colored), and were submitted to qPCR analysis for selected genes (Figure 2). All the three analyzed genes related to the primary metabolism were induced in BS rachis after BS berry symptoms became visible (Figure 2a–c): asparagine synthetase (VviASN1; *VIT_06s0004g06830*), galactinol synthase (VviGOLS3; *VIT_14s0060g00810*), and stachyose synthase (VviSTAS1; *VIT_07s0005g01680*). In addition, a very similar expression profile was observed for the transcription factor ethylene-responsive factor 3 (VviERF003; *VIT_09s0002g09140,* data not shown).

The cell wall biosynthetic genes (cellulose synthases) tested by qPCR were more than 2-fold decreased in T2, in line with the negative implication of this group in GSA analysis. The detailed picture at the level of individual genes may be a bit more complicated, as shown by qPCR, as we see both CESA4 (*VIT_07s0005g04110*) (Figure 2d) being moderately enhanced *(**t*-test, *q* < 5%) and CESA8 (*VIT_10s0003g01560*) (Figure 2e) being repressed in BS rachis samples (*q* < 1%), with the repression especially strong before veraison during berry growth phase I. On the other hand, the expression of EXPA6 (*VIT_06s0004g04860*) (Figure 2f) started to increase at veraison (EL-35) in BS rachis and was significantly enhanced in EL36/1 and EL36/2 (*q* < 5%). The cell wall hydrolytic enzymes xyloglucan endotransglucosylase/hydrolase 32 (VviXTH32; *VIT_06s0061g00550*), β-D-xylosidase (VviBXL1; *VIT_05s0077g01280*), and pectin methylesterase 3 (VviPME3; *VIT_09s0002g00320*) (Figure 2g–i) were highly induced in BS rachis after BS symptoms developed (EL-36/1) (*q* < 0.1%).

In summary, according to our expression profile analyses, cellulose synthesis genes are active in the rachis during cluster growth, yet we determined a reduced activity in BS rachis. Additionally, several pectinesterases and xyloglucan endotransglucosylase/hydrolases (XETs, XTHs) were repressed, especially in T2. With visible BS symptoms at T2, genes of the category “hydrolytic enzymes of polysaccharides” were enhanced in expression, together with the highly induced genes BXL1, XTH32, EXPA6, and PME3, while individual genes in other categories were partly up- and partly down-regulated, leaving the roles of these genes in BS symptom development open.

### 2.2. Berry Cell Wall Modifications Induced by the Disorder

Due to the strong relevance of the cell wall metabolism observed in the genes modulated in the rachis, we dug in our previous dataset performed on pre-symptomatic and symptomatic berries [49] in order to verify if there was a modulation of this specific metabolism also in the fruit. Therefore, we retrieved all the DEGs related to cell wall biosynthesis, degradation, and modification, including the hydrolytic enzymes.

In pre-symptomatic green berries (EL-33), only a cell wall degradation pectinesterase was enhanced in BS fruits (Appendix A). It is interesting to consider that this was the only DEG found in that developmental stage [49]. In the following sampling (EL-35, veraison), the DEGs involved in the cell wall metabolism modulated by the berry shrivel disorder incremented, representing 7.7% of the total number of DEGs in that developmental stage. In particular, there were three genes enhanced, while the other 20 were repressed (Appendix A, Figure 3). The five top genes showing the highest absolute degree of variation (in RPKM) in EL-35 between BS-vs-H were genes all down-regulated belonging to the major category of cell wall modification and degradation; in fact, they were a: (i) invertase/pectin methylesterase inhibitor, (ii) xyloglucan endotransglucosylase/hydrolase 32 (iii) polygalacturonase inhibiting protein PGIP1, (iv) invertase/pectin methylesterase inhibitor, and (v) expansin B04. To notice that four out of five are listed as SWITCH genes [49], meaning that in normal development, these genes should start to be highly expressed during the second phase of ripening.

With the progression of the symptoms, which from these developmental stages onward were visible on the berries, more genes were differentially expressed; in the EL-36/1, 85 genes (5.7% of the total DEGs) were modulated with 55 genes induced and 30 lower expressed in BS berries (Appendix A, Figure 3). Furthermore, in the EL-36/2, 139 genes (5.7% of the total DEGs) resulted in DEGs with 63 genes enhanced and 76 repressed (Appendix A, Figure 3).

Interestingly, only four genes were commonly modulated in EL-35, EL-36/1, and EL-36/2 in the berries; three were annotated as pectin acetylesterase, and they were all lower expressed in BS rachis in the three stages, while the fourth gene was a β-1,3 glucanase enhanced in EL-35 and further increased in EL-36/1/2.

## 3. Discussion

Herein, we report on substantial transcriptional changes related to cell wall modification and cell wall degradation in the rachis and berries of grape clusters showing the symptoms of berry shrivel.

By analyzing the expression in the rachis of pre-symptomatic and symptomatic grape clusters, we found indications of two different modification patterns in relation to the grape cluster growth curve. Early in development, cellulose synthases are expressed at a high level in the rachis of both sample organs. In BS samples, the GSA analysis of all samples and the DEGs for samples at T2 identified significantly reduced expression, which was also confirmed by qPCR for CESA8.

Cell walls are highly complex, and cellulose is a major structural component synthesized at the plasma membrane by cellulose synthase complexes consisting of six rosette subunits formed by multiple isoforms of cellulose synthase (CESA) enzymes. In contrast, cellulose synthase-like (Csl) enzymes synthesize the non-cellulosic polysaccharide components of the cell wall [55,56]. The traditional model of plant cell growth in surface area is associated with cell wall loosening, also called wall stress relaxation, and by the viscoelastic extension driven by water uptake without the necessary addition of new wall polymers, although this addition is needed to maintain wall integrity [57]. Nevertheless, grape clusters grow fast from flowering towards veraison in approximately fifty days [25], underlining the necessity to build stable cell wall structures to support berry weight enhancement. A specific lack of cellulose synthase, especially early in grape cluster development as indicated by CESA8 (Figure 2), CESA2, and GSLG3 (Appendix A) expression at T2, could weaken this important structural function of the rachis and make cell walls more prone to degradation. Cell wall biosynthesis could also be modulated by a large number of repressed xyloglucan endotransglucosylase/hydrolase genes, acting as XET/XTHs in wall assembly and cell growth by breaking and re-joining hemicellulose chains [58]. The specific functional characterization of this gene family in *Vitis* is pending.

In a previous study, we could show that pedicels of BS-affected grape clusters were thinner, which may indicate a reduced growth with an impact on assimilate transport [47]. On the other hand, no obvious major anatomical rearrangements were observed in the vascular tissue organization in the rachis and pedicels of BS grape clusters with light microscopy [45], while a localized cell wall thickening of the secondary phloem in the rachis and pedicels of BS grapes, as well as degraded and collapsed cells near cambium cells, were observed [45]. All these observations point towards cell wall modification and degradation in BS symptomatic grape rachis, for example, due to the activity and altered expression of EXPA6 (*VIT_06s0004g04860*), XTH32 (*VIT_06s0061g00550*), BXL1 (*VIT_05s0077g01280*), and PME3 (*VIT_09s0002g00320*), confirmed by qPCR. Moreover, the GO enrichment analyses congruently reported several GO groups enriched in T1 or in T2 BS rachis related to these processes (e.g., cell wall organization, xyloglucan and hemicellulose metabolic process, etc. (Figure 1)). A previous study observed alternating bands of secondary hard and soft phloem in peduncles of BS affected [44] which could be an indication of senescence or a protection from mechanical compression due to strong tensile stress [45]. In our case, cell wall thickening and cell wall modifications point towards a local strengthening of cell walls to protect the cambium and thereby sustain cell growth. In parallel, cell wall degradation and modification of cell wall polysaccharides may occur to either support cell wall relaxation for growth or protection against abiotic stresses.

In general, modifications of cell wall polysaccharides are associated with the de-polymerization of homogalacturonan and the loss of neutral sugars (galactose or arabinose) from hairy regions of pectins (rhamnogalacturonan I and II) accompanied by the action of cell wall hydrolases such as β-galactosidases (β-Gals), pectin methylesterases (PMEs), polygalacturonases (PGs), and pectate lyases (PLs) [59,60,61]. The expression of these genes represents the second pattern of expression profile in BS rachis, where we saw several members of these categories highly induced in the rachis of BS clusters (T2). In addition to EXPA6, XTH32, and BXL1, several hydrolytic enzymes were induced in T2 BS rachis, e.g., one β-Gals, one PMEs, two PGs, and one PLs. At the same time, other genes were repressed, especially the gene family xyloglucan endotransglucosylase/hydrolases and pectinesterases, leaving the question of the consequences of the cell wall modifications process open.

Expansins enable cell expansion by a pH-dependent non-enzymatical relaxation of the cell wall [62]. Among the tested genes with qPCR, EXPA6 seems to be earlier induced at EL-35 in BS rachis compared to the other tested genes. Although the mode of action of expansins is not yet fully elucidated, they are thought to act on the hydrogen bonds linking cellulose and hemicellulose, especially xyloglucan, thereby allowing their sliding to each other and promoting cell expansion [63,64]. According to the expression profiles of the grapevine gene family of expansins (see Figure 3a in [65]), EXPA6 is normally higher expressed in rachis tissue during fruit set (FS) and post-fruit set (PFS), in active green tissues such as in buds at the bud burst and after it, in young and well-developed inflorescences, tendrils, and green stems. The higher expression in BS rachis after veraison could hint towards an immature cluster development earlier in development. The high expression of EXPA6 and XTH32 in BS rachis at T2 and confirmed by qPCR could indicate an enhanced process of cell wall relaxation, which could facilitate the access of hydrolytic enzymes to cell wall polymers. During grape berry ripening, it has been shown that xyloglucan endotransglucosylase/hydrolase (XTH) acts early before the activity of enzymes leading to pectin polymerization [27]. The repression of several members of this gene family in T2 suggests a disturbed cell wall assembly and loosening capacity, as most xyloglucan endotransglucosylase/hydrolase act as endotransglucosylases [58].

PMEs de-esterify methyl-esterified D-galactosiduronic acid units in pectin, which becomes easier degraded by polygalacturonase [66]. The action of PMEs can lead to the formation of free carboxylic groups, which, in the presence of calcium ions, cause the precipitation of pectin due to the formation of calcium pectate [67]. We analyzed the expression of PME3, induced in BS rachis at EL-36/1. By analyzing sieve plates with SEM in BS-affected rachis sections, we observed a carbohydrate-based net type material covering the entire sieve plate [45], a structure that was not present in samples from healthy grape clusters. Confirmation would be needed, but this could be a consequence of the action of PMEs to facilitate calcium pectate precipitation.

The β-D-xylosidase 1 (BXL1) is among the highest up-regulated genes in our study, an enzyme supposed to have as substrate glucuronoarabinoxylan (GAX) and loosening its interaction in secondary cell wall formation, e.g., during vascular development [68]. Interestingly BXL1 is documented as responsive to sugar starvation together with BGAL4 in *Arabidopsis* [69,70], suggesting that cell walls may function as a reserve of carbon under sugar starving conditions [70]. There is no annotation for BGAL4 in the current grapevine genome, but among the up-regulated genes at T2, there is one β-galactosidase BGAL1 (*VIT_11s0016g02200*). The most consistent symptom of BS in grape berries is a stop in sugar accumulation, and its consequence for rachis and pedicels sugar support is unknown. Similarly to the study in *Arabidopsis* [70], we determined a reduced expression of genes involved in cell wall biosynthesis (cellulose synthase, xyloglucan endotransglycosylase) at T2, as well as genes of different secondary metabolism pathways (MYBPA1 (*VIT_15s0046g00170*, flavonoid pathway), HMGR (*VIT_04s0044g01740*, terpenoid pathway), laccase (*VIT_08s0007g01910*, *VIT_13s0067g01970*, simple phenolic pathway), and CHS2 (*VIT_14s0068g00930*, phenylpropanoid pathway and F3’5’H d (*VIT_06s0009g02840*, flavonoid pathway)). In summary, cellulose synthase, xyloglucan endotransglycosylase, and genes involved in the secondary metabolism are reduced in the rachis of BS grapes, a profile similar to sugar starvation, while genes related to cell wall modification and cell wall degradation are induced at T2. In parallel, several studies observe callose deposition at the sieve plates suggesting a reduced phloem conductance [45,52,71]. Our microarray study did not detect callose synthase genes being significantly higher expressed as |log_2_FC| > 1 at any sampling timepoint, which could point towards an earlier induction or a non-transcriptional regulated process.

Among the highest induced genes in BS rachis at T2, there were the primary metabolism genes ASN1 (*VIT_06s0004g06830*), GOLS3 (*VIT_14s0060g00810*), and STAS1 (*VIT_07s0005g01680*). Galactinol synthase and stachyose synthase support the production of higher oligosaccharides in plants, which are often accumulated as plant stress responses, e.g., drought, salinity, or cold stress, with function in membrane stabilization and scavenging of reactive oxygen species [72,73]. Similarly, ASN1 is one of the highest expressed genes in BS T2, with a log_2_FC of 5.3. It has also been reported to be induced by abiotic stress, specifically osmotic and salt stress [74]. The gene itself is a key enzyme in the biosynthesis of the nitrogen-rich amino acid, asparagine, cycling nitrogen within the plant [75]. Therefore, the induction of these genes could point towards a stress response in BS rachis, potentially caused by osmotic stress or ROS imbalance.

In grape berries, cell wall modification is integral for berry softening at the onset of berry ripening and to regain berry growth during the period of sugar accumulation [32,63,76]. During grape berry ripening, the action of cell wall modifying enzymes has been documented, e.g., expansins, xyloglucan endotransglycosylase/hydrolases, β-galactosidases, and polygalacturonases, as well as pectin methyl esterases and pectate lyases [27,33,63,77,78]. Recently, we reported on altered transcriptional profiles in BS berries along with a timescale from pre-symptomatic to symptomatic samples [49]. Interestingly, we found no metabolic alterations in the berry transcriptome during the first growth phase of berries (EL-32, EL-34), also not related to cell wall biosynthesis and cell wall modification, as shown herein. At veraison and before the BS symptoms were visible (EL-35), several genes of the cell wall modification and degradation categories were lower expressed in BS berries. In contrast, later during berry ripening and with BS berries showing symptoms, we observed both enhanced and repressed genes of the same categories. This suppression of these categories in BS berries at veraison is not in temporal accordance with the observed suppression of cellulose biosynthesis and xyloglucan endotransglycosylase/hydrolases in BS rachis, where we observed the down-regulation of genes at T2 after detectable symptoms. Reasons could be manifold, e.g., the different years of sampling, the pooling of samples for the microarray study, or simply that molecular processes are coordinated differently in rachis and berry organs.

Interestingly only a few genes showed the same expression pattern in BS berries at EL-35, EL-36/1, and EL-36/2. Among them, three pectin acetylesterases were lower expressed. Pectin acetylesterase cleaves ester bonds between cell wall polysaccharides and thereby releases acetate. Reports indicate that changes in *O*-acetylation affect processes like photomorphogenesis and defense response [79]. Thereby pectin acetylesterase regulates the status of pectin acetylation with consequences on the capacity of remodeling of cell wall polysaccharides, which determines their extensibility [80]. Therefore, one could speculate that cell wall extensibility based on pectin modification is reduced in BS berries and in BS rachis with several genes of the pectinesterase family repressed in the more advanced development stages with visible disorders symptoms. Cell growth could be limited by less extensible cell walls preventing the establishment of a growing sink tissue, but on the other hand, these observed expression patterns could be the consequence of earlier modified processes. Cell wall modification and degradation could result in cell degradation and the loss of cell compartmentation, which has been observed as enhanced cell death late during berry development [81] and in association with late-season dehydration and BS [82,83,84]. Recently, FDA staining revealed that cell death in BS berries precedes cell death in the BS rachis, with more severe symptoms near the cluster tip [46]. If, how, and to which extent cell wall modification and degradation observed in our study via expression analyses and the loss in cell viability determined microscopically are linked needs to be further studied, as well as their contribution to BS induction.

## 4. Materials and Methods

### 4.1. Plant Material Sampling

Grape berries and rachis samples were collected from grape clusters of the red grape cultivar Blauer Zweigelt (*Vitis vinifera* L.; grafted on rootstock Kober 5BB; planted in 1974) in 2011 and 2013 from a commercial vineyard located in Lower Austria (Antlasberg, Mailberg). Details on pedo-climatic conditions and viticultural management have been previously described [49,54,85]. Grape clusters (*N* = 300) in both years were randomly labeled within the vineyard at EL-31 (BBCH-75), corresponding to pea-sized berries. In order to obtain berries, rachis, and pedicels from pre-symptomatic grape clusters and not disturb the ripening process of the labeled clusters, each cluster was sampled only once by collecting the distal part of the grape clusters (including around ten berries, pedicels, and rachis) with a scalpel. All samples were immediately frozen in liquid nitrogen in the field and stored in the laboratory at −80 °C. Time series of samples (six individual sampling timepoints) were collected in both years, ranging from 30 to 75 days after anthesis, which corresponds from EL-33 (BBCH-79) until EL-38 (BBCH-89) in 2011 and EL-32 (BBCH-77) until EL-37 (BBCH-89) in 2013. Before harvesting at the end of the season, all labeled grape clusters were categorized into healthy (H, control) or BS-affected (BS) ones according to a visual evaluation and measurement of soluble solids (°Brix) [49,54]. Follow-up sorting of frozen samples collected from EL-32 and EL-38 allowed the analyses of BS pre-symptomatic and H pre-veraison samples in comparison collected at the same sampling date as well as healthy ripening and BS symptomatic grape clusters later during berry ripening. Veraison is stated at the onset of berry coloring at around 53 days after anthesis (DAA) in 2011 and 55 DAA in 2013, which corresponds to EL-35. The first BS symptoms were observed approximately 7–10 days later at around 62 DAA in 2011 and 65 DAA in 2013 at EL-36/1 [54]. Expression analyses were performed with three biological replicates. These replicates resulted from three pooled vineyard samples of the same sampling timepoint and category (H, healthy; BS berry shrivel symptoms). In total, about 25–30% of sampled grape clusters developed BS symptoms, which overestimates the BS incidence in the vineyard by about 5–10% as soon as symptoms were visible a more targeted sampling took place. Frozen plant material was ground to a fine powder under liquid nitrogen using a ball mill (Retsch M400) ahead of RNA extraction.

### 4.2. RNA Extraction and cDNA Preparation for qPCR

RNA extraction from rachis tissue was performed with a modified CTAB protocol with lithium chloride (0.33 vol of 8 M LiCl) precipitation as previously described [85,86] from 100 mg of frozen ground plant material. Quality and quantity control of total RNA was performed with a NanoDrop 2000c UV-vis Spectrophotometer (Thermo Scientific, Wilmington, DE, USA), followed by cDNA preparation of 1 µg total RNA with the QuantiTect Reverse Transcription Kit (Qiagen, Hilden, Germany) according to the manufacturer’s recommendation. Total RNA from grape berries for RNASeq analyses was extracted with the “Spectrum Plant total RNA” kit (Sigma-Aldrich, St. Louis, MO, USA) from 200 mg frozen berry powder. RNA quality control, mRNA library preparation, and sequencing have been previously described [49]. A summary of the samples and tissues used for transcriptional expression analyses presented in the current study is given in Table 2.

### 4.3. Microarray Analyses of Rachis Tissue and qPCR

Rachis tissue from pre-symptomatic grape clusters (T1: collected samples combined from 49 DAA and 55 DAA, EL-34,35, 2011) and symptomatic grape clusters (T2: collected samples combined from 69 DAA and 75 DAA, EL-36,37, 2011) was used to systematically profile gene expression with a custom-made Agilent SurePrint Custom GE 4x44 microarray (Agilent Protocol G2514F-031062) previously described [87]. In short: labeling and hybridization were performed with the Two-Color Microarray-Based Gene Expression Analysis—Quick Amp Labelling kit with the Tecan HS Pro Hybridization protocol (V 5.7, May 2008; Agilent, Santa Clara, CA, USA): Double-stranded Cy3-labeled and Cy5-labeled cDNA was synthesized from 50 ng of total RNA using a T7-oligo(dT) primer. Fragmentation and washing were performed as previously described [87]. Probes were hybridized to the microarrays for 17 h at 65 °C in a Tecan HS 4800 Pro Hybridization Station (Tecan, Männedorf, Switzerland), applying a dye-swap configuration in order to minimize dye bias. Microarrays were scanned with an Agilent G2505C scanner at a resolution of 5 µm in double-pass mode, with the green and red dye channel lasers set to 100% power, yielding 20-bit TIFF images, which were further analyzed with Agilent Feature Extraction Software version 10.10.1.1 using default parameters (Agilent Protocol GE2_1010_sep10). Raw signal estimates from image analysis were analyzed using the R package “limma” [88]. After normalization, robust multi-chip models were fit using the lmFit function [89]. Obtained q-values were corrected for multiple testing following Benjamini–Hochberg to control the false discovery rate [90]. For the statistical tests, individual gene variances were moderated using an Empirical Bayes approach that drew strength from transferring variance characteristics from the set of all genes in the test for each individual gene [89]. We correctly account for intra-chip replicate probe correlations [91] and add robustness from weights from an iterative weighted least-squares fit. Genes were finally selected that exhibited significantly stronger fold change than a threshold, |log_2_(fold change)| > 1 [92]. Statistics were computed for comparisons of infected tissues vs. healthy controls at each of the two time points T1 and T2.

To complement GeneOntology enrichment analyses, we performed gene set analysis (GSA) [93] for a selection of gene sets of interest. All berry shrivel samples were compared to their healthy controls pairwise. The gene sets were selected based on the working hypotheses of disturbed cell wall metabolism, and are listed in Appendix A. We report the Benjamini–Hochberg-corrected false discovery rate.

The genes regulated significantly more than 2-fold based on our microarray results at T2 were the basis for selecting genes of potential relevance for testing our hypothesis of the relevance of cell wall biosynthesis, cell wall modification, and cell wall degradation for further analysis by qPCR in a higher-resolution time course of berry development. Samples collected at six sampling dates in 2013 ranging from EL-32 to EL-36 were analyzed (Table 2). qPCR analyses were performed using the Rotor-Gene Q cycler (Qiagen, Hilden, Germany) with the KAPA SYBR Fast qPCR universal kit (Sigma-Aldrich, St. Louis, MO, USA) in a final volume of 12 µL. Cycling conditions were as follows: 4 min at 95 °C, 40 cycles for 8 s at 95 °C, 20 s at 60 °C, 30 s at 72 °C, and 5 s at 75 °C with fluorescence measurement. Analyses were performed in duplicates, and results are expressed as normalized expression values (NRQs) using the reference genes actin (*VIT_04s0044g00580*) and ubiquitin (*VIT_16s0098g01190*) for normalization [94]. A list of primers used is given in Appendix A.

### 4.4. RNASeq Analyses of Grape Berries

Description of the RNASeq analyses and presentation of the results (GO enrichment analyses, DEG of sugar and anthocyanin metabolism, and SWITCH genes) have been described in a previous study [49]. Here, we present the information related to cell wall modifications during berry development and berry ripening compared to control and BS-affected grape clusters.

### 4.5. Statistical Analyses

All statistical comparisons were conducted using IBM SPSS Statistics 21 or R. The statistical analysis of the microarray data is detailed above. For qPCR, significant differences were tested by comparing H and BS-affected berries with Student’s *t*-test (*p* < 0.05) if the normal distribution was ensured; otherwise, non-parametric Mann–Whitney-U tests were conducted. Functional GO enrichment analysis was performed with the tool g:Profiler using the g:GOSt functional profiling tool with the Benjamini–Hochberg-corrected false discovery rate [95]. Heatmaps were drawn using the R package “gplots”.

## 5. Conclusions

In conclusion and as a summary of our study, we propose the hypothesis of reduced cell growth and cell wall biosynthesis during the first berry growth phase in BS rachis, which does not result in substantial anatomical reorganization of the vascular system but may affect the strength of cell walls. Similarly, in berries at the onset of ripening, we observed a reduction of cell wall modifying enzymes along with other SWITCH genes, potentially leading to a delayed regain of berry growth after the lag phase in parallel to the already determined delay in sugar accumulation and ripening-related processes [49]. In both tissues, genes of the categories “cell wall biosynthesis”, “cell wall modification”, “cell wall degradation”, and “hydrolytic enzymes of polysaccharides” were modulated in expression in symptomatic BS grape clusters. Specifically pronounced is the enhanced expression of hydrolytic enzymes in BS rachis and the high expression of genes BXL1, XTH32, and EXPA6 at EL-36/1 and EL-36/2 in BS rachis as determined with qPCR, while the highest modulated cell wall modifying and degradation genes in BS berries were down-regulated. The question of a coordinated regulation in both tissues could not be answered at this point. Future dedicated studies with carefully matched samples will need to target this knowledge gap, including the timing of cell wall modification and/or cell wall reorganization in both tissues. The major challenge will be the linkage between cell wall assembly and modification in BS rachis with its ability to transport assimilates towards ripening berries and the feedback of berry metabolism (e.g., sink strength) on pedicel and rachis development, including vascular tissue development. A method specifically inducing BS would help to investigate the induced processes in different tissues, but, currently, no reliable method is available. Nevertheless, our results and first insights could form a valuable contribution to establishing a BS induction method in the near future.

## Figures and Tables

**Figure 1 plants-11-02159-f001:**
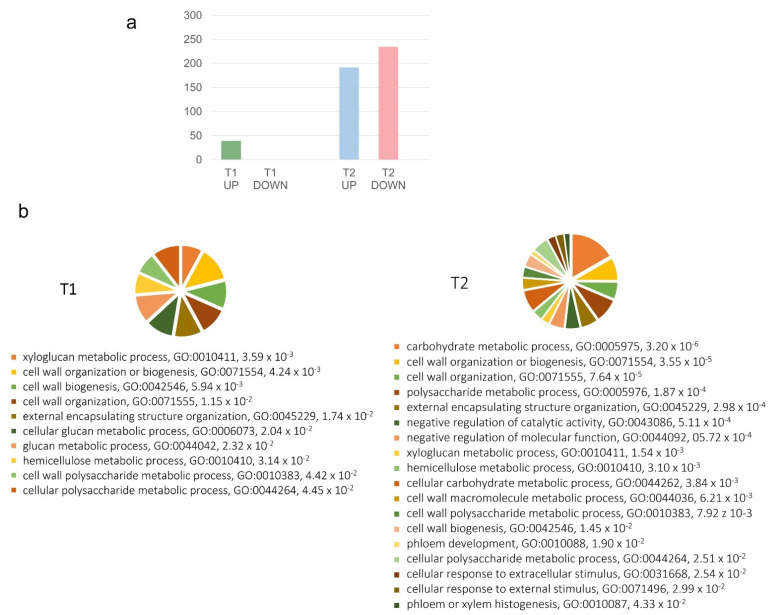
(**a**): Number of genes significantly more than 2-fold differentially expressed in symptomatic vs. non-symptomatic (BS/H) rachis tissues in the T1 and the T2 developmental stages; (**b**): Biological Process functional enriched categories in T1 and in T2; the name of the category with its GO ID number are reported next to the *q*-values giving the false discovery rates after Benjamini–Hochberg correction.

**Figure 2 plants-11-02159-f002:**
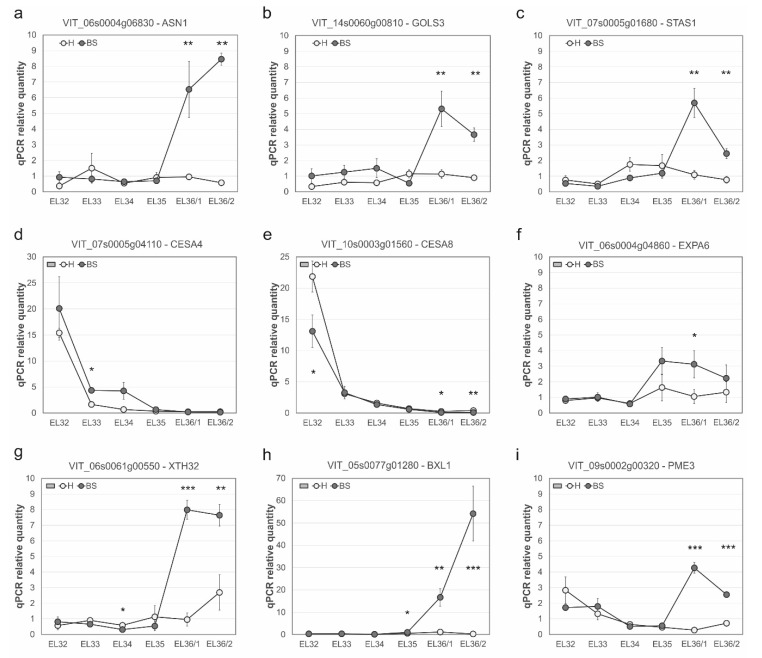
Relative expression of selected genes determined by qPCR related to the primary metabolism and cell wall modifications. NRQs are presented for healthy (H) and berry shrivel (BS) affected grape clusters in the rachis. Samples were collected at six sampling dates throughout the berry development and berry ripening in 2013 (EL-32, EL-33, EL-34, EL-35, EL-36/1, EL-36/2). (**a**) Asparagine synthetase (VviASN1; *VIT_06s0004g06830*), (**b**) galactinol synthase (VviGOLS3; *VIT_14s0060g00810*), (**c**) stachyose synthase (VviSTAS1; *VIT_07s0005g01680*), (**d**) cellulose synthase (VviCESA4; *VIT_07s0005g04110*), (**e**) cellulose synthase (VviCESA8; *VIT_10s0003g01560*), (**f**) expansin A (VviEXPA06; *VIT_06s0004g04860*), (**g**) xyloglucan endotransglucosylase/hydrolase 32 (VviXTH32; *VIT_06s0061g00550*), (**h**) β-D-xylosidase (VviBXL1; *VIT_05s0077g01280*), and (**i**) pectin methylesterase 3 (VviPME3; *VIT_09s0002g00320*). Data represent mean values ± standard error (N = 3). Statistically significant differences were tested with a *t*-test and are indicated with an asterisk (*** 0.001; ** 0.01, * 0.05).

**Figure 3 plants-11-02159-f003:**
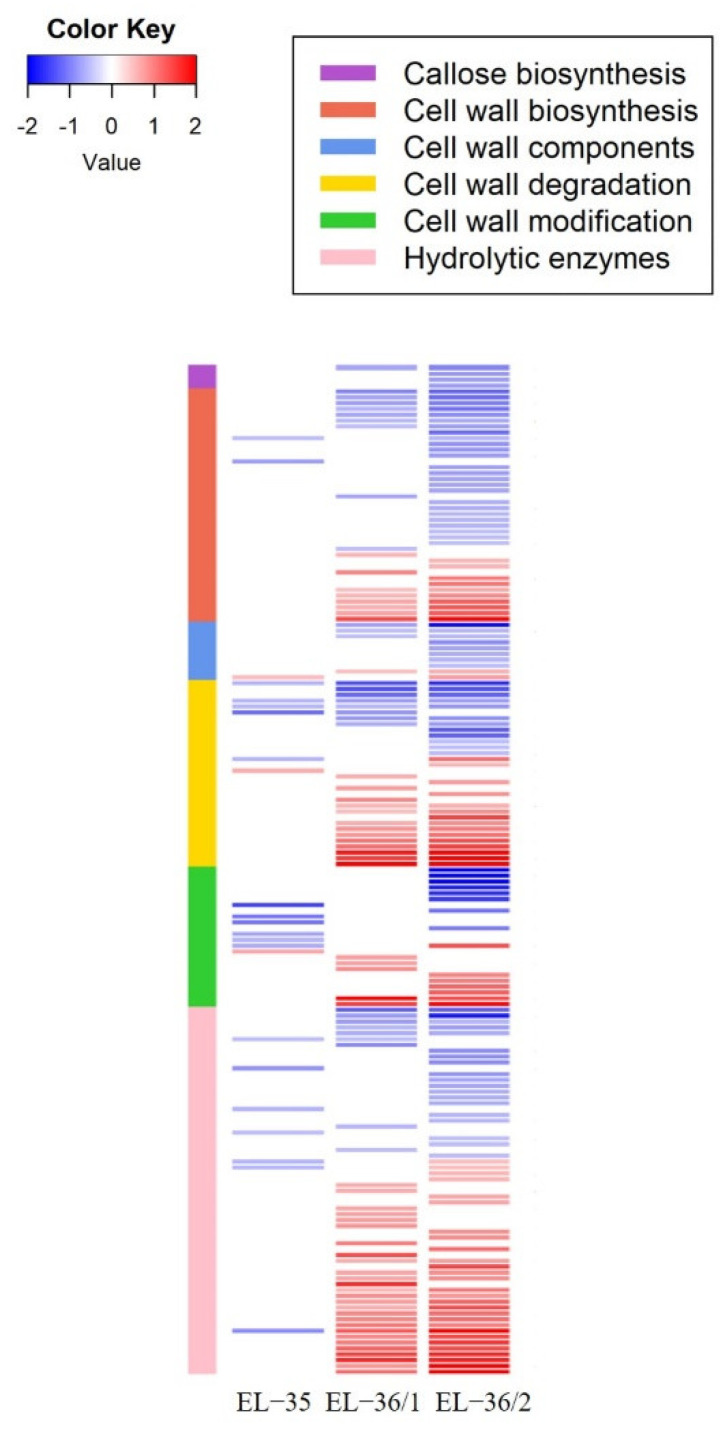
Genes differentially expressed during the progression of the berry shrivel symptoms in ripening berries belonging to the cell wall metabolism are represented in a heatmap. Values are presented as the log_2_FC (BS/H) at EL-35, EL-36/1, and EL-36/2. Blue and red colors indicate down- and up-regulation in BS, respectively. A white cell denotes the non-significance. The colored sidebar on the left indicates their functional groups.

**Table 1 plants-11-02159-t001:** List of selected genes differentially expressed related to the cell wall metabolism in the rachis of BS grape clusters at stage T1 (pre-symptomatic EL-34,35, 2011) and at stage T2 (symptomatic EL-36,37, 2011). Results are shown as log_2_ FC and significance is based on an adjusted *q*-value of 0.05; significant results in bold letters. Further information is given in Appendix A.

Functional Group	12Xv1 ID	logFC T1	*q*-Value	Rank	Log FC T2	*q*-Value	Rank	Annotation
Cell wall biosynthesis	VIT_02s0025g01750	−0.39	n.s.		**−1.35**	6.6 × 10^−3^	384	Cellulose synthase CSLG3
	VIT_04s0008g02830	0.21	n.s.		**1.83**	2.4 × 10^−8^	237	Galactokinase like protein
	VIT_07s0005g04110	0.54	n.s.		**−1.30**	3.7 × 10^−2^	422	Cellulose synthase CESA4
	VIT_14s0083g01100	0.58	n.s.		**1.49**	2.6 × 10^−6^	290	Alpha-1,4-glucan-protein synthase 1
	VIT_16s0039g02020	**1.41**	2.1 × 10^−3^	33	−0.69	n.s.		Cellulose synthase CSLD3
	VIT_18s0122g00120	0.05	n.s.		**−1.81**	5.7 × 10^−13^	158	Cellulose synthase CESA2
Cell wall degradation	VIT_04s0044g01000	0.21	n.s.		**−1.34**	3.2 × 10^−2^	418	Pectinesterase family
	VIT_05s0077g01280	0.49	n.s.		**4.60**	1.4 × 10^−42^	3	Glycosyl hydrolase family 3-β-xylosidase BXL1
	VIT_08s0007g04820	0.84	n.s.		**1.51**	1.7 × 10^−4^	340	Pectate lyase
	VIT_08s0007g07690	0.40	n.s.		**1.57**	1.4 × 10^−6^	283	Polygalacturonase inhibiting protein PGIP1
	VIT_08s0007g07880	0.31	n.s.		**1.30**	1.4 × 10^−2^	395	Polygalacturonase GH28
	VIT_09s0002g00320	0.81	n.s.		**1.20**	2.2 × 10^−2^	408	Pectinesterase PME3
	VIT_11s0016g03020	−0.20	n.s.		**−1.42**	3.6 × 10^−6^	294	Pectinesterase family
	VIT_16s0050g01110	0.36	n.s.		**1.90**	1.2 × 10^−12^	165	Polygalacturonase GH28
	VIT_16s0098g01900	−0.12	n.s.		**−1.35**	8.4 × 10^−8^	247	Pectinesterase family
Cell wall modification	VIT_00s0323g00050	0.03	n.s.		**1.53**	5.4 × 10^−10^	205	Invertase/pectin methylesterase inhibitor
	VIT_00s0386g00050	0.44	n.s.		**−2.26**	3.3 × 10^−17^	105	XET/XTH
	VIT_01s0026g00200	0.22	n.s.		**−2.25**	1.2 × 10^−22^	68	Xyloglucan endotransglucosylase/hydrolase 28
	VIT_03s0088g00650	0.48	n.s.		**2.44**	1.8 × 10^−25^	51	Xyloglucan/xyloglucosyl transferase
	VIT_05s0062g00480	0.28	n.s.		**−1.66**	3.7 × 10^−5^	324	Xyloglucan endo-transglycosylase
	VIT_06s0004g04860	**2.01**	3.6 × 10^−18^	10	**1.71**	7.9 × 10^−13^	160	Expansin A (VvEXPA06)
	VIT_06s0061g00550	**1.47**	5.3 × 10^−8^	20	**2.88**	3.5 × 10^−33^	22	[SWITCH] XTH 32
	VIT_11s0052g01180	**1.60**	3.5 × 10^−8^	19	−0.53	n.s.		XET/XTH
	VIT_11s0052g01250	0.04	n.s.		**−1.87**	5.3 × 10^−17^	108	XET/XTH
	VIT_11s0052g01270	**1.42**	2.6 × 10^−3^	34	−0.86	n.s.		XET 6
	VIT_11s0052g01280	**2.03**	1.4 × 10^−12^	12	−0.92	n.s.		XET/XTH
	VIT_12s0134g00160	0.14	n.s.		**−2.59**	2.2 × 10^−26^	48	XET/XTH
Hydrolytic enzyme	VIT_00s0455g00040	0.23	n.s.		**−1.39**	1.7 × 10^−2^	402	Glycosyl transferase family 8 protein
	VIT_03s0017g02240	0.28	n.s.		**1.56**	2.7 × 10^−7^	264	Endo-1,3-β-glucosidase precursor
	VIT_03s0180g00280	-0.07	n.s.		**2.50**	2.1 × 10^−20^	79	Indole-3-acetate β-glucosyltransferase
	VIT_05s0062g00310	1.11	n.s.		**−1.66**	9.3 × 10^−9^	231	UDP-glucoronosyl/UDP-glucosyl transferase UGT75C1
	VIT_06s0004g01430	0.04	n.s.		**2.39**	1.0 × 10^−20^	76	ABA glucosidase
	VIT_06s0004g07230	0.30	n.s.		**3.20**	2.8 × 10^−33^	21	Indole-3-acetate β-glucosyltransferase
	VIT_06s0061g00120	−0.39	n.s.		**2.68**	2.5 × 10^−28^	38	β-1,3-glucanase
	VIT_08s0007g06040	−0.06	n.s.		**3.30**	1.2 × 10^−38^	7	[SWITCH] β-1,3-glucanase
	VIT_08s0040g01470	−0.18	n.s.		**1.66**	6.7 × 10^−9^	229	Cis-zeatin O-β-D-glucosyltransferase
	VIT_11s0016g02200	0.74	n.s.		**3.98**	4.8 × 10^−36^	13	β-galactosidase
	VIT_12s0028g00050	0.49	n.s.		**1.29**	1.5 × 10^−3^	362	β-1,3 glucanase
	VIT_18s0001g06090	0.88	n.s.		**−1.40**	4.7 × 10^−5^	326	Cis-zeatin O-β-D-glucosyltransferase
	VIT_19s0014g03240	−0.05	n.s.		**−1.49**	1.3 × 10^−5^	312	β-mannosidase 4

**Table 2 plants-11-02159-t002:** Summary of samples used for expression analyses. Control (healthy) samples and BS-affected (BS) samples were analyzed in comparison at each sampling time point. All analyses were performed with three biological replicates.

Analyses Performed	Year of Sampling	Grape Cluster Tissue	Description	DAA	EL-Scale	Sampling Date (DD/MM/YYYY)
Microarray	2011	rachis	Pre-symptomatic T1	49 and 55	EL-34 and EL-35	29 July 2011 & 4 August 2011
Microarray	2011	rachis	Symptomatic T2	69 and 75	EL-36 and EL-37	17 August 2011 & 24 August 2011
qPCR	2013	rachis	Pre-symptomatic	30	EL-32	11 July 2013
qPCR	2013	rachis	Pre-symptomatic	44	EL-33	25 July 2013
qPCR	2013	rachis	Pre-symptomatic	51	EL-34	1 August 2013
qPCR	2013	rachis	Veraison, pre-symptomatic	58	EL-35	8 August 2013
qPCR	2013	rachis	Symptomatic	65	EL-36/1	15 August 2013
qPCR	2013	rachis	Symptomatic	72	EL-36/2	22 August 2013
RNASeq	2013	berries	Pre-symptomatic	30	EL-32	11 July 2013
RNASeq	2013	berries	Pre-symptomatic	44	EL-33	25 July 2013
RNASeq	2013	berries	Pre-symptomatic	51	EL-34	1 August 2013
RNASeq	2013	berries	Veraison, pre-symptomatic	58	EL-35	8 August 2013
RNASeq	2013	berries	Symptomatic	65	EL-36/1	15 August 2013
RNASeq	2013	berries	Symptomatic	72	EL-36/2	22 August 2013

## Data Availability

Micorarray grape rachis: array design is available at A-MTAB-534—Vitis1agl180v1_1378399567999; data supporting the results of this article have been deposited in the Array Express database at EMBL-EBI (www.ebi.ac.uk/arrayexpress) under accession No E-MTAB-12113 on the 16 August 2022). RNASeq data of grape berries: All raw transcriptomics reads have been deposited in NCBI Sequence Read Archive (http://www.ncbi.nlm.nih.gov/sra). The BioProject and SRA accession are PRJNA436693 and SRP134067, respectively.

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
