# Peer review of "Expression Analyses in the Rachis Hint towards Major Cell Wall Modifications in Grape Clusters Showing Berry Shrivel Symptoms"

_plants, 2022, doi:10.3390/plants11162159_

Round 1
Reviewer 1 Report
Manuscript deals with a very current topic from the point of view of viticultural research. The authors follow up on their previous articles, which is clearly visible in the text. The work is based on new research methods. The overall level is very high in all parts of the mauscript. The interpretation of the results is clear and understandable. I propose publication without changes.
Author Response
Thank you very much for your encouraging feedback.
Reviewer 2 Report
Dear Authors,
That’s great to know your research on berry shrivels correlated to the changes in cell wall metabolism at the transcriptional level. The paper is very readable. I want to offer some comments in hopes you will reconsider the following lines.
Line 42: ‘double sigmoid growth curve,’ I recommend further explaining the term and adding reference.
Line 86: you might need to provide more information to explain the correlation between berry shrivel and early sugar accumulation disorder.
Line 286: that’s interesting ‘cell wall thickening of the secondary phloem’; however, you might need to explain further why ‘thickening of phloem’ is a symptom of ‘cell wall modification’ at plant physiological or metabolic levels.
Line 459: I’m curious about how many percent of berries in one cluster showing berry shrivel symptoms were considered BS samples?
Author Response
Line 42: ‘double sigmoid growth curve,’ I recommend further explaining the term and adding reference.
Reference added.
Line 86: you might need to provide more information to explain the correlation between berry shrivel and early sugar accumulation disorder.
Actually, these both terms refer to the same phenomenon. Berry shrivel is very often used as well for late season dehydration, therefore we wanted to clearly state that we are taking about a ripening disorder which start early during grape berry ripening. We tried to clarify this by modifying the sentence: Sunburn, late season dehydration (LSD), bunch stem necrosis (BSN), and an early sugar accumulation disorder called berry shrivel (BS) are among the grapevine ripening disorders that substantially impact grape yield and berry quality.
Line 286: that’s interesting ‘cell wall thickening of the secondary phloem’; however, you might need to explain further why ‘thickening of phloem’ is a symptom of ‘cell wall modification’ at plant physiological or metabolic levels.
Indeed, this aspect was not well explained. Two sentences were added to the discussion to contribute to the current knowledge. Nevertheless, a more specific answer is not possible as cell wall compounds analyses are missing especially on a cell compartment level.
Line 459: I’m curious about how many percent of berries in one cluster showing berry shrivel symptoms were considered BS samples?
All berries of a BS grape cluster were considered as BS samples. With BS, complete grape clusters are affected, the symptom severity (shrinking of berries) may be different at the end of the ripening process, but our samples did no show shrinking symptoms yet. They are lower in soluble solids and different in color and turgor. Shrinking berries appear later. Considering all sampled clusters, we observed about 25-30 BS clusters in 2011-2013. We did add one sentence to Materials and Methods providing this information.